# Obstetric Violence Is Prevalent in Routine Maternity Care: A Cross-Sectional Study of Obstetric Violence and Its Associated Factors among Pregnant Women in Sri Lanka’s Colombo District

**DOI:** 10.3390/ijerph19169997

**Published:** 2022-08-13

**Authors:** Dinusha Perera, Muzrif Munas, Katarina Swahnberg, Kumudu Wijewardene, Jennifer J. Infanti

**Affiliations:** 1Family Health Bureau, Ministry of Health, Colombo 01000, Sri Lanka; 2National Cancer Control Programme, Ministry of Health, Colombo 00500, Sri Lanka; 3Department of Health and Caring Sciences, Faculty of Health and Life Sciences, Linnaeus University, SE-391 82 Kalmar, Sweden; 4Department of Community Medicine, Faculty of Medical Sciences, University of Sri Jayewardenepura, Gangodawila, Nugegoda 10250, Sri Lanka; 5Department of Public Health and Nursing, Faculty of Medicine and Health Sciences, NTNU-Norwegian University of Science and Technology, NO-7491 Trondheim, Norway

**Keywords:** obstetric violence, domestic violence, maternity care, Sri Lanka, prevalence study, vulnerable populations

## Abstract

The phenomenon of obstetric violence has been documented widely in maternity care settings worldwide, with scholars arguing that it is a persistent, common, but preventable impediment to attaining dignified health care. However, gaps remain in understanding local expressions of the phenomenon, associations with other types of violence against women, and implications for women’s trust and confidence in health providers and services. We focused on these issues in this cross-sectional study of 1314 women in Sri Lanka’s Colombo district. Specifically, in this study, we used Sinhalese and Tamil translations of the NorVold Abuse Questionnaire and the Abuse Assessment Screen to measure prevalence of women’s experiences with obstetric violence in maternity care and lifetime and pregnancy-specific domestic violence. Then, the results were interpreted by considering the women’s sociodemographic characteristics, such as age, ethnicity, and family income, to reveal previously undocumented associations between obstetric and domestic violence during pregnancy, as well as other factors associated with experiencing obstetric violence. We argue that obstetric violence is prevalent in government-sector (public) maternity care facilities in the Colombo district and is associated with young age, lower family income, non-majority ethnicity, and rural residency. Significantly, this study sheds light on a serious concern that has been underexamined, wherein women who report experiencing obstetric violence are also less likely to be asked by a health care provider about domestic violence experiences. Further research at the clinical level needs to focus on appropriate training and interventions to ensure women’s safety and cultivate relationships between patients and health care providers characterized by trust, confidence, and respect.

## 1. Introduction

Women experiencing violence or other abuse, described with terms that include wife battering, gender-based violence (GBV), violence against women (VAW), domestic violence (DV), and intimate partner violence (IPV), have been studied worldwide for about five decades, since the 1970s [1,2,3,4]. Prevalence rates vary widely across contexts, reflecting both diversity in actual occurrence of violence, as well as researchers’ definitions and measurement instruments. However, specialized agencies responsible for international public health have recognized violence or other abuse specifically targeting women as a widespread and devastating global public health problem [5]. Since the 1990s, patients in health care, particularly female patients, who have been experiencing violence—i.e., abuse, neglect, disrespect, or mistreatment—in health care facilities have been examined through the concepts of patient satisfaction and dissatisfaction, with the aim of improving patient care. However, growing evidence suggests that using the satisfaction typology only reflects what one is asking for; that is, patients report more satisfaction when asked specifically about “patient satisfaction” than if asked about their dissatisfaction [6,7]. This might explain why the emergent issue of women experiencing violence in health care, particularly in maternity health care facilities—also referred to as obstetric violence—has long been silenced or neglected worldwide.

Obstetric violence is difficult to define due to the complexity and variety of its manifestations and local expressions. The World Health Organization (WHO) conceptualizes it in terms of abuse, disrespect, and mistreatment in childbirth that result in violations of women’s dignity by health professionals, ranging from outright physical abuse to humiliation caused by verbal abuse or lack of confidentiality to neglect that results in unnecessary pain and avoidable complications [8]. Obstetric violence often focuses on labor and childbirth even when referring to maternity care, which includes pregnancy, given that these are moments in which women are particularly vulnerable to health care abuse and over-medicalization, or non-medically justified obstetric interventions, e.g., episiotomy and caesarean section. Other important components of obstetric violence are dehumanization and non-consensual care, as well as overall conversion of biological processes into pathological ones [9,10,11]. It has been argued that obstetric violence needs to be analyzed separately from other kinds of medical violence because it is a feminist issue: gendered violence perpetrated on healthy women, and a type of violence that can be viewed as a form of sexual violence [10].

With continuing policy and legal structure changes, attention and awareness raised toward human rights and women’s fundamental rights, and the general evolution of human rights perspectives, obstetric violence in health care has captured growing attention worldwide over the past two decades. According to internationally adopted human rights and standards, obstetric violence can amount to a violation of women’s fundamental human rights [12,13,14]. Recognizing the issue’s significance, the White Ribbon Alliance for Safe Motherhood in 2011 declared seven universal rights for childbearing women: 1. freedom from harm and ill-treatment; 2. the right to information, informed consent, and respect for choices and preferences, including the right to companionship of their choice wherever possible; 3. confidentiality and privacy; 4. dignity and respect; 5. equality, freedom from discrimination, and equitable care; 6. the right to timely health care and the highest attainable level of health; and 7. the right to liberty, autonomy, self-determination, and freedom from arbitrary detention [15]. In 2015, the WHO released a statement emphasizing that “every woman has the right to the highest attainable standard of health, which includes the right to dignified, respectful health care throughout pregnancy and childbirth, as well as the right to be free from violence and discrimination” [8]. Venezuela was the first country (2007) to define obstetric violence formally as a punishable form of violence against women, with more Latin and South American countries following suit [16]. Furthermore, punishable acts under VAW laws have been amended in Argentina (2009) and Mexico (2014) to address obstetric violence [17].

The first documented reports of abusive behaviors among health care providers in obstetric care were byproducts of investigating health-seeking behaviors [18], which is not unusual. Afterward, an emergent body of qualitative and quantitative research outlining different typologies of obstetric violence recognized that many women experience this form of violence in health facilities worldwide. Generally, violence that health care providers perpetrate has been studied under the rubrics of the terms patient satisfaction [19] and patient dissatisfaction [20], ethical transgressions among staff [21,22], disrespect and abuse [23,24,25], abuse in health care [26], and obstetric violence [10,27]. A literature review conducted by Virginia and Arachu in 2017 included a total of 57 research articles that measured patient mistreatment during childbirth (i.e., disrespect, obstetric violence, and other abuse), particularly within Latin America and the Caribbean [28]. These studies provided ample scientific evidence that women experience violent or disrespectful experiences in maternity health care settings worldwide. Of the 65 studies included in Bohren et al.’s systematic 2015 review on mistreatment of women during childbirth in health facilities globally, obstetric violence had been reported in several studies from Sub-Saharan Africa, Oceania, Europe, the Middle East, North Africa, North America, and Latin America. Furthermore, two studies examined two countries in South Asia: India and Bangladesh. Three studies from Tanzania and Nigeria quantified that 15% to as many as 98% of women experience at least one form of mistreatment—e.g., physical abuse, sexual abuse, and/or verbal abuse—during childbirth [29]. In 2019, Jungari et al. examined 11 studies related to disrespectful maternity care practices during childbirth in India, including mistreatment and abuse. The prevalence of disrespect and abuse ranged from 10 to 77.3% in India, and it was evident that these adverse experiences create negative impacts on utilization of health facilities for childbirth, establishing a psychological barrier between women and health care providers [30].

Sri Lanka is a developing country in the South Asian region with good public health indicators—stronger than many other countries in the region and other developing countries worldwide, and comparable with some developed countries that have nearly 10 times its gross domestic product [31]. To achieve this success, Sri Lanka’s accessible and high-quality education system and universal access to free health care have contributed immensely. Maternal and child health care services are provided optimally through a well-established, unique public health care system that includes domiciliary care, as well as institutional care. For at least one decade now, in every year, more than 95% of pregnant women are registered, more than 94% of them have visited field prenatal clinics, and almost all the deliveries (99.9%) take place in medical facilities [32]. Although Sri Lanka did not reach the target of 23 maternal deaths per 100,000 live births by 2015, as set in the Millennium Development Goals, a considerable decline in maternal mortality, from 92 maternal deaths per 100,000 live births down to 36, was achieved during the 1990–2015 period [33].

However, among all these highlighted and un-highlighted victories in maternal health in Sri Lanka, issues with space in facilities to provide optimal quality care need to be emphasized. In less than 100 public hospitals where specialist obstetric care is available, nearly 320,000 childbirths take place per year [34]. This space inadequacy in maternity wards and labor rooms, which highly affects patients’ privacy; a shortage of health care professionals; and extremely busy units are evident in most of these institutions. Less than 800 medical officers/additional medical officers of health and around 6000 public health midwives provide care for nearly 350,000 pregnant women at the field level annually [32]. Previous studies have found that mistreatment and violence can occur during women’s interaction with their health care providers, as well as through systemic failures at the health facility and health care system levels [10,29]. With Sri Lanka’s existing limitations, maternity care settings have become vulnerable spaces where women potentially can experience obstetric violence.

Knowledge about obstetric violence in Sri Lanka is limited, and research is scarce, although our study team has been documenting some occurrences. In 2015, Infanti et al. reported obstetric violence in a qualitative study on pregnant women who recounted past experiences of violence that health care providers perpetrated, particularly during childbirth, in public hospitals [35]. We also described accounts of obstetric violence in a study published in 2018—i.e., emotional, physical, and sexual violence that health care providers perpetrated during women’s present and past pregnancies [36]. Based on our 2018 paper’s recommendations, Swahnberg et al. conducted a pilot intervention to assess the possibilities of using a participatory theatre technique as a method to reduce and prevent obstetric violence in Sri Lanka [37]. An overwhelming need exists to develop scientific evidence further to account for and intervene more effectively to prevent and mitigate obstetric violence in Sri Lanka and, by extension, upgrade and optimize maternity care quality. Moreover, pregnancy is viewed as a window of opportunity to identify women living with DV [38]. Obstetric care, including prenatal and postpartum care, is a significant arena in which to prevent DV. However, the mere possibility of obstetric violence occurring might dissuade women from seeking maternity health care altogether [39,40].

This is the first quantitative study to date in Sri Lanka to estimate the prevalence of obstetric violence and its associated factors. Furthermore, the study aimed to investigate the potential association between DV and experiencing obstetric violence, as well as any significant relationships between the occurrence of obstetric violence and disclosure of DV among pregnant women in Sri Lanka.

## 2. Materials and Methods

The data were collected for this cross-sectional study from April 2014 to January 2015 in Sri Lanka’s Colombo district, the nation’s administrative and commercial capital. People in this region live in highly urbanized and developed areas, rural and highly populated underdeveloped urban areas, and a plantation estate sector. All ethnic and religious groups in the nation are represented in the Colombo district, with the majority belonging to the Sinhala ethnicity and Buddhist religion (the majority ethnic and religious groups in Sri Lanka, respectively), whereas the minorities comprise Muslim and Tamil ethnicities and Hindu, Catholic, and Islam religions. Vast socioeconomic and sociocultural differences exist among the population across the district. Eligible study participants included pregnant women attending maternity care in all types of health facilities in Colombo district, 16 years and older, with at least one prior birth experience. Women were not eligible to participate in the study if intellectual disability or speech impairments precluded them from giving informed consent or from answering verbally administered interview questions, or if they had critical health concerns requiring specialist care or hospital admission during their current pregnancy.

The required sample size for this study was 1375 participants, which was calculated using a formula for cross-sectional studies of qualitative variables (see [41]), adjusted for a presumed non-response rate of 10%, and based on a predicted prevalence rate of obstetric violence encountered during previous pregnancies of 50% given prior studies in the country on other types of gender-based violence experienced by pregnant women. A potential loss of precision in estimating population prevalence, due to cluster sampling, was minimized using a correction technique advocated by Bennett et al. and based on knowledge from other studies in Sri Lanka of homogeneity in the study setting [42,43].

Ultimately, 1375 women were recruited to the study in a two-stage sampling process aimed to ensure accurate representation of the district’s diverse population. Pregnancy registration is a mandatory duty of public health midwives (PHMs), making them the field-level maternal and child health care providers in Sri Lanka [32]. Thus, in the initial stage of recruitment to the study, a list was generated of all PHM areas in the district. The PHM areas are the smallest health administrative areas in the country and were our primary ‘sampling units’ or ‘clusters’. Of the total PHM areas, we applied computer-generated number tables to randomly select 55 for sampling based on the assumption that a minimum of 25 participants would need to be included from each cluster to achieve the desired sample size and minimize bias. Subsequently, in a second sampling stage, the study’s inclusion and exclusion criteria were applied, excluding all primipara women. Then, we randomly selected 25 pregnant women from each sampling unit by applying computer-generated random numbers to the regularly updated pregnancy registers of the 55 sampling units (PHM areas).

Three main instruments were used for data collection. First, following a thorough literature review, the Abuse in Health Care (AHC) section of the NorVold Abuse Questionnaire (NorAQ) was selected to measure obstetric violence [44]. Both obstetric violence and experiences with AHC comprise emotional, physical, sexual, and verbal abuse [7,21,29]. Obstetric violence, which also includes a patient’s subjective experiences, becomes a similar phenomenon to AHC, as the two have critical attributes in common: these patients feel like they lose their value as human beings and experience health care encounters devoid of care [45]. As such, emotional, physical, and sexual violence that health care providers perpetrate during immediate past pregnancy could be measured through questions related to AHC in the NorAQ questionnaire. External translators translated these questions from English into Sinhalese and Tamil, the two major local dialects in Sri Lanka. The translated instrument was assessed for cultural adaptation, face and content validity using a modified Delphi technique and a multidisciplinary panel of experts. Second, for participants who answered that they had experienced obstetric violence during their immediate past pregnancies, the study team developed a set of additional follow-up questions to gain more in-depth information than what the AHC questions yielded in the NorAQ. Namely, participants were asked when in the past pregnancy the violence had occurred, the type of perpetrator, place of incident, current suffering, and impact on health seeking behaviors and other outcomes attributed to the experience. Finally, DV was examined using a modified version of the Abuse Assessment Screen (AAS) [46]. The AAS was originally a five-item questionnaire, and the study team developed and added two more questions, asking women whether a health care worker had “ever asked them about DV,” and among those reporting DV, the women were asked whether they had “ever disclosed about DV” to a health care worker. External translators translated the AAS instrument from English to Sinhalese and Tamil, and the same multidisciplinary panel of experts mentioned above assessed the instrument for clarity and comprehensibility.

Two female sociology graduates who were conversant in the two main local languages were selected and trained as data collectors (research assistants). The high literacy rate of females in Sri Lanka enabled self-completion of the questionnaires, but the research assistants administered the AAS verbally to maintain this screening instrument’s original methodology. Informed written consent was obtained from each of the participants following the provision of verbal and written study information. Data collection took place in prenatal clinics in the field, where it was possible to ensure privacy, confidentiality, and safety. Following data collection, the participants were offered contact and referral information to *Mithuru Piyasa*, which are hospital-based sexual- and gender-based violence care centers that provide services to violence survivors. The Ethical Review Committee of the Faculty of Medical Sciences, University of Sri Jayewardenepura, Sri Lanka, reviewed all study methods, instruments, and procedures (Ref. No. 8/14).

For analytical purposes, we focused on the binary indicator of “obstetric violence” vs. “no obstetric violence” as the main outcome variable. At least one reported experience with obstetric violence during immediate past pregnancy care—i.e., during the prenatal, intrapartum, or postpartum (up to 42 days) periods, with an identified health care provider perpetrator—was counted as obstetric violence. We conducted the statistical analyses for prevalence estimates and multivariate analysis using logistic regression to identify the adjusted correlates for obstetric violence using the Statistical Package for Social Sciences (SPSS) (Version 21, IBM, New York, NY, USA). First, associations between obstetric violence and background characteristics were tested at the univariate level, then statistically significant factors (*p* < 0.05) were included in a logistic regression model.

## 3. Results

Out of the 1375 eligible pregnant women invited to complete the questionnaire, 95.56% (1314) participated. Their mean age was 31.39 (SD 5.22), and the majority (73.6%) belonged to the Sinhalese ethnic group. More than half (59.3%) had enrolled in or completed 11 years (secondary level) of school education, and about 79.4% were housewives (unemployed). More than 90% received prenatal care from government sector health facilities, and a nearly identical proportion gave birth in a government hospital.

The results affirm that women receiving obstetric care experience violence that health care providers perpetrate in government health care facilities. The women encountered various forms of violence assessed in the questionnaire, with the majority exposed to emotional violence (Table 1). Even though questions on different types of obstetric violence were presented separately, the same women could be subjected to more than one type of violence. Among the women, during their immediate past pregnancy, 18.1% (238) experienced obstetric violence that health care providers perpetrated, with almost all the women (235) classifying it as a type of emotional obstetric violence. Of the remaining participants, 0.8% (11 women) reported physical violence, and 0.2% (two women) reported sexual violence that a health care provider perpetrated.

Most of the participants’ background characteristics differed significantly (*p* < 0.05) between the two groups, i.e., those who experienced obstetric violence vs. the “no obstetric violence” group. The women’s ages, education levels, ethnicities, monthly family incomes, and areas of residence were all significantly different among the two groups. The health care facilities where women received prenatal care and the institutions where they underwent childbirth, as well as mode of birth, also were all significantly different between the two groups. However, employment status, number of living children, and partner’s education and employment type were not significantly different between the groups (Table 2).

The data collected using the AAS instrument regarding women’s DV experiences also were examined to search for any potentially significant differences between the two groups. The prevalence of women’s experiences with current and lifetime DV, and DV in pregnancy, was significantly different (*p* < 0.001) between the “obstetric violence” and “no obstetric violence” groups (Table 3). Even though asking pregnant women about DV is a mandate for PHMs in Sri Lanka, fewer women who reported obstetric violence than those who did not had been asked about DV by a health care provider (*p* = 0.003). Disclosure of DV was examined only among women who reported lifetime DV, and disclosure was not significantly different between the two groups (Table 3).

To identify the correlates of obstetric violence, the characteristics initially compared at the univariate level and the statistically significant correlates (*p* < 0.05) were included in a logistic regression model. Some of the participants’ sociodemographic characteristics were significant correlates of obstetric violence. Ages 16–21 (OR 3.29, CI 1.28–4.46), education level above the secondary level (OR 1.85, CI 1.29–2.65), belonging to the Muslim ethnicity (OR 3.10, CI 2.00–4.18), low income (OR 3.68, CI 1.61–8.40), and living in a rural area (OR 1.95, CI 1.40–2.72) were significant correlates of obstetric violence when adjusted for all included variables (Table 4).

Several characteristics of obstetric care and facilities also were significant correlates of obstetric violence (Table 4); for example, receiving prenatal care in a government health facility (OR 3.27, CI 1.53–7.02), childbirth occurring in a government hospital (OR 2.51, CI 1.18–5.34), and vaginal mode of delivery (OR 1.60, CI 1.05–2.45) were significant correlates of obstetric violence. Notably, experiencing DV during pregnancy correlated with obstetric violence, with a high odds ratio (OR) (OR 7.45, CI 3.87–14.32).

## 4. Discussion

As in our previously published research [36,37], the present study’s findings indicated that obstetric violence affects a significant proportion of women receiving maternity care in government health institutions in Sri Lanka’s Colombo district. Some groups of women appear to be more vulnerable to obstetric violence due to particular characteristics or experiences, such as young age, more formal education, lower family income, non-majority ethnicity, and rural dwelling status. These findings are similar to those reported in other countries (see, e.g., [29]). However, much of the scientific literature on this topic takes a compartmentalized approach to vulnerability and “victimization” [47], i.e., it reports on obstetric violence experienced by women of low socioeconomic status, women with disabilities, immigrant women, black women, or other women of color [48,49,50]. The present study’s findings draw attention to the fact that one woman can experience several of these intersecting exposures or circumstances simultaneously, resulting in complex and cumulative discrimination and “othering.” These particularly vulnerable women’s violence experiences must be better documented and addressed in health provider training and other interventions to ensure more equitable health care systems.

The present study’s findings also demonstrate a strong interrelationship between experiencing obstetric violence and past and present DV. This reinforces other studies (see, e.g., [51,52]) that have identified correlations between experiencing several types of violence, e.g., childhood abuse, witnessing parental violence, and violence that intimate partners perpetrate. Significantly, the present study sheds light on a concern, underexamined to date, wherein women who report experiencing obstetric violence are also less likely to be asked by a health care provider about any DV—despite training and a professional mandate to inquire or “screen” for DV during prenatal care. This may indicate that some women are so stigmatized by multiple and intersecting factors, such as their ethnicity, cumulative violence experiences, and social class, that maternity health care providers fail to recognize or otherwise overlook their suffering. Alternatively, these women’s complex health needs may be too overwhelming or difficult for overworked, fatigued, and under-resourced health providers to address, or mistreatment of these women is normalized or believed to be acceptable in institutional settings characterized by overwork, stress, and other constraints [27]. The study’s findings indicate that obstetric violence is not just isolated acts, but rather is embedded in context-specific norms of class, race, gender, medical power, and hierarchy. These are areas that require attention among researchers, hospital and health care system managers, educators, and other involved stakeholders to ensure that women attain optimal maternity care and effective formal help from the health care sector.

As in all studies collecting retrospective data, this study’s findings are limited by participant’s abilities to recall incidents of obstetric violence in health care in relation to immediate past pregnancies. For some women, this was more than 10 years before participating in the current study. Their ability to recall these events accurately would be variable and some may have ‘blocked’ such painful memories even subconsciously. Relatedly, we asked the women for information regarding the perpetrators (health care workers). Some of them were unable to recall the type of health provider and we excluded these responses from the analyses to be cautious, but this may contribute to under-reporting of the phenomenon. Furthermore, under-reporting of obstetric violence by the women is to be expected due to the sensitive nature of the problem and the fact that a health care provider (first author) was the primary interviewer. Despite these limitations, the findings of this study have constructive implications for Sri Lanka’s healthcare system.

Although efforts have been taken to ensure stronger public and community health responses to address DV [53], the study provides more data that maternity health care providers still require training on how to inquire about DV respectfully and ensure women’s rights to dignified, non-discriminatory health care. Health care providers also need to be trained to recognize and address their own experiences with violence and oppression, as well as potentially discriminating and harmful attitudes, and to understand the mechanisms of obstetric and interpersonal/domestic violence. Studies incorporating assessments of obstetric violence prevention and intervention measures particularly tailored to women experiencing multiple forms of adversity, violence, marginalization, and stigmatization are a positive direction forward for facilitating improved maternal health outcomes. Theoretical research also is needed to develop more sophisticated conceptualizations of obstetric violence wherein multiple power modalities are acknowledged [27]. Moreover, more studies on obstetric violence are needed that focus on risk and protective influences of community and social context.

## 5. Conclusions

By estimating the prevalence of and associated factors for experiencing obstetric violence in public maternity care settings in Sri Lanka, as well as associations with past or present experiences with DV, this study exposes violence as a routine phenomenon in women’s lives and a threat to the health and well-being of pregnant women, new mothers, their children, and families. Prenatal and postpartum care are potential environments for effectively identifying and supporting women living with or vulnerable to violence. However, this study’s findings indicate that the youngest, most economically disadvantaged women, particularly those who have experienced DV, and regardless of higher education, are also among the most vulnerable groups for abuse and disrespect at the hands of their health care providers in maternity care. This can serve as a warning to the health care system concerning social and economic stratification’s pervasive effects on dignified treatment in health care settings. Care comprises technical, practical, and emotional components, and Sri Lanka has achieved significant indicators of technical success in the provision of maternal health care. However, more work remains to ensure that care also meets women’s needs in terms of the right to feel safe and supported, and to receive respectful and appropriate care.

## Figures and Tables

**Table 1 ijerph-19-09997-t001:** Prevalence of obstetric violence.

Event	N	% (95% CI)
Experienced violence involving health care providers during immediate past pregnancy care (obstetric violence)	238	18.1 (16.02–20.18)
Experienced emotional type of obstetric violence (neglected, insulted, verbally abused, etc.)	235	17.8 (15.73–19.87)
Experienced physical obstetric violence (hitting, slapping, pushing, etc.)	11	0.8 (0.32–1.28)
Experienced sexual obstetric violence	2	0.2 (−0.04–0.44)

**Table 2 ijerph-19-09997-t002:** Background characteristics of pregnant women reporting obstetric violence.

	Total (N = 1314)	Obstetric Violence *		*p*-Value
		Yes: n (%)	No: n (%)	
**Age**				
16–21	32	13 (40.62)	19 (59.38)	χ^2^ = 11.22
22–35	999	176 (17.62)	823 (82.38)	df = 2
36–44	283	49 (17.31)	234 (82.69)	*p* = **0.004**
**Education level**				
Grades 0–5	144	35 (24.31)	109 (75.69)	χ^2^ = 7.21
Grades 6–11	766	122 (15.93)	644 (84.07)	Df = 2
>Grade 12	404	81 (20.05)	323 (79.95)	*p* = **0.027**
**Ethnicity**				
Sinhala	967	160 (16.55)	807 (83.45)	χ^2^ = 15.13
Tamil	166	27 (16.27)	139 (83.73)	df = 3
Muslim	170	49 (28.82)	121 (71.18)	*p* = **0.001**
**Living children**				
0–1 child	829	155 (18.70)	674 (81.30)	χ^2^ = 0.519
2 children	379	65 (17.15)	314 (82.85)	df = 2
≥3 children	106	18 (16.98)	88 (83.02)	*p* = 0.771
**Employment status**				χ^2^ = 1.57
Employed	271	42 (15.50)	229 (84.50)	df = 1
Housewife	1043	196 (18.79)	847 (81.21)	*p* = 0.21
**Family income/month**				
Up to LKR 30,000	60	17 (28.33)	43 (71.67)	χ^2^ = 6.28
LKR 30,001–60,000	1014	186 (18.34)	828 (81.66)	df = 2
LKR 60,001 and above	240	35 (14.58)	205 (85.42)	*p* = **0.043**
**Living area**				χ^2^ = 5.93
Rural	521	111 (21.31)	410 (78.69)	df = 1
Urban	793	127 (16.02)	666 (83.98)	*p* = **0.015**
**Prenatal care**				χ^2^ = 9.07
Government sector	1184	227 (19.17)	957 (80.83)	df = 1
Private sector only	126	10 (7.94)	116 (92.06)	*p* = **0.002**
**Place of delivery**				χ^2^ = 8.04
Government hospital	1190	227 (19.08)	963 (78.66)	df = 1
Private hospital	110	9 (8.18)	101 (91.82)	*p* = **0.005**
**Mode of delivery**				χ^2^ **=** 7.13
Vaginal delivery	1043	204 (19.56)	839 (80.44)	df = 1
Cesarean section	271	34 (12.55)	237 (22.0)	*p* = **0.008**
**Partner’s education**				χ^2^ **=** 0.34
Up to grade 11	890	160 (18.0)	730 (82.0)	df = 1
12 and above	424	78 (18.4)	346 (81.6)	*p* = 0.854
**Partner’s job category**				
Professional	172	24 (14.5)	147 (85.5)	χ^2^ = 1.76
Non-professional	1103	202 (18.3)	901 (81.7)	df = 1
Unemployed	16	2 (12.5)	14 (87.5)	*p* = 0.415

***** At least one reported experience with obstetric violence by a health care provider during immediate past pregnancy care was counted as obstetric violence. Notes: Internal dropouts 0–2%; bold *p*-Values indicate evidence of statistical significance.

**Table 3 ijerph-19-09997-t003:** Domestic violence compared with obstetric violence experiences.

	Total (N = 1314)	Obstetric Violence *		*p*-Value
		Yes: n (%)	No: n (%)	
**Ever experience DV during pregnancy?**				χ^2^ = 49.89
Yes	53	29 (54.70)	24 (45.30)	df = 1
No	1261	209 (16.60)	1052 (83.40)	*p* < **0.001**
**Lifetime DV (AAS)? ^a^**				χ^2^ = 20.01
Yes	414	104 (25.10)	310 (74.90)	df = 1
No	900	134 (14.90)	766 (85.10)	*p* < **0.001**
**Current DV (AAS)? ^b^**				χ^2^ = 14.70
Yes	132	40 (30.30)	92 (69.70)	df = 1
No	1182	198 (16.80)	984 (83.20)	*p* < **0.001**
**Ever asked by health care provider about DV?**				χ^2^ = 8.72
Yes	726	152 (20.90)	574 (79.10)	df = 1
No	588	86 (14.60)	502 (85.40)	*p* = **0.003**
**Ever disclose DV ^c^ to a health care provider?**				χ^2^ = 0.004
Yes	55	14 (25.50)	41 (74.50)	df = 1
No	359	90 (25.1)	269 (74.90)	*p* = 0.951

* At least one reported experience with obstetric violence from a health care provider during immediate past pregnancy care was counted as *obstetric violence*. Note: bold *p*-Values indicate evidence of statistical significance. ^a^ Experiencing DV-lifetime: If the woman ever has been emotionally or physically abused and/or physically abused during the last year and/or physically abused during pregnancy and/or sexually abused last year in AAS, these were combined and labeled “Experience of DV-lifetime”. ^b^ Experiencing DV-current: If the woman ever has been physically abused last year and/or sexually abused last year in AAS, these were combined and labeled “Experience of DV-current”. ^c^ Only 414 women who reported DV during their lifetime were asked about DV.

**Table 4 ijerph-19-09997-t004:** Crude and adjusted odds ratio (OR) for obstetric violence * and associated factors adjusted for all included variables.

		OR_Crude_	CI (95%)	OR_Adjusted_	CI (95%)
Participant Characteristics					
**Age**	16–21 years	**3.27**	**1.51–7.06**	**3.29**	**1.28–4.46**
	22–35 years	1.02	0.72–1.45	0.96	0.69–1.39
	36–44 years	1		1	
**Education ^a^**	Low	**1.70**	**1.11–2.60**	1.26	0.76–2.08
	High	1.32	0.97–1.81	**1.85**	**1.29–2.65**
	Middle	1		1	
**Ethnicity**	Tamil	0.98	0.63–1.53	1.01	0.60–1.68
	Muslim	**2.04**	**1.41–2.97**	**3.10**	**2.00–4.81**
	Sinhalese	1		1	
**Income ^b^**	Low	**2.33**	**1.19–4.51**	**3.68**	**1.61–8.40**
	Middle	1.32	0.89–1.95	1.21	0.78–1.88
	High	1		1	
**Living area**	Rural	**1.42**	**1.07–1.89**	**1.95**	**1.40–2.72**
	Urban	1		1	
**Prenatal care**	Government institutions	**2.75**	**1.41–5.33**	**3.27**	**1.53–7.02**
	Private only	1		1	
**Mode of delivery**	Vaginal delivery	**1.70**	**1.15–2.51**	**1.60**	**1.05–2.45**
	Cesarean section	1		1	
**Place of delivery**	Government hospital	**2.65**	**1.32–5.31**	**2.51**	**1.18–5.34**
	Private hospital	1		1	
**Experiencing DV during pregnancy**	Yes	**6.08**	**3.47–10.66**	**7.45**	**3.87–14.32**
	No	1		1	

* At least one reported experience with obstetric violence by a health care provider during immediate past pregnancy care was counted as obstetric violence. Note: bold *p*-Values indicate evidence of statistical significance. ^a^ Education: low (no schooling and primary-level school education only); middle (above primary up to secondary-level school education); or high (above secondary-level school education). ^b^ Income: low (up to 30,000); middle (30,001 to 60,000); or high (60,001 and up) in LKR per month. Note: Internal dropouts 0–2%.

## Data Availability

Data is available upon request of co-authors D.P. or M.M. due to privacy/ethical restrictions on publication in a public domain.

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
