# Peer review of "Obstetric Violence Is Prevalent in Routine Maternity Care: A Cross-Sectional Study of Obstetric Violence and Its Associated Factors among Pregnant Women in Sri Lanka’s Colombo District"

_ijerph, 2022, doi:10.3390/ijerph19169997_

Round 1

Reviewer 1 Report

It was my pleasure to review this manuscript dealing with obstetric violence in Sri Lanka.

Obstetric violence is a serious problem that affects many women around the world. Unfortunately it is a subject that has been silent and I think it is time to make it visible, call it by its name and face it once and for all to try to reduce it.

I congratulate the authors for choosing such an interesting as well as controversial topic for the medical community.

In order to improve the quality of the manuscript I am going to make a series of comments:

1 Introduction: I think it would be necessary to introduce the topic by giving a definition of obstetric violence. For this, what the WHO and the UN say about it can be taken as a reference.

2 material and methods:

I think it would be necessary to clearly describe the criteria for inclusion and exclusion of the sample.

For analytical purposes, the authors focused on the binary indicator of “obstetric violence” versus “no obstetric violence” as the main outcome variable. It would be interesting to explain which zone variables are involved in classifying the patient in one group or another.

3 Results. I liked how the results were treated. Without a doubt, the strong point of the manuscript is the logistic regression study.

4 Discussion. In the discussion it is necessary to express clearly and concisely what the limitations of the study were.

Otherwise I found the manuscript quite good and interesting.

Kind Regards

Author Response

AUTHORS’ RESPONSES TO REVIEWER 1
First, we would like to thank you for the generous and helpful review of our manuscript. We have amended the manuscript following your input and we have detailed some of our considerations during this process below (our responses in red font).

REVIEWER 1
1 Introduction: I think it would be necessary to introduce the topic by giving a definition of obstetric violence. For this, what the WHO and the UN say about it can be taken as a reference. Yes, thank you for this comment which we have now addressed in the second paragraph of the manuscript (lines 64-69). It certainly improves the manuscript to be clearer about the terminology of obstetric violence despite its contestation/debate. 
2 Material and methods: I think it would be necessary to clearly describe the criteria for inclusion and exclusion of the sample. Again, thank you for pointing out the lack of clarity in this respect in the original version of the manuscript. The study’s inclusion/exclusion criteria are now specified in the revised manuscript (end of first paragraph in the ‘Materials and Methods’ section, lines 184-190).
For analytical purposes, the authors focused on the binary indicator of “obstetric violence” versus “no obstetric violence” as the main outcome variable. It would be interesting to explain which zone variables are involved in classifying the patient in one group or another. In the analyses, at least one reported experience with obstetric violence by a health care provider during immediate past pregnancy care was counted as obstetric violence (line 275). We have now added information to the revised manuscript about the additional follow-up questions the study team developed for women who reported experiencing obstetric violence (see lines 241-247). Furthermore, given the sensitivity of the topic in the context, if the research participant did not also identify a perpetrator type (i.e., type of health care provider), we decided to exclude this from the count as obstetric violence to be very cautious. This is now clearer in the manuscript text (line 268) and the discussion of study limitations (393-397). 
3 Results. I liked how the results were treated. Without a doubt, the strong point of the manuscript is the logistic regression study. Thank you very much for this generous praise.
4 Discussion. In the discussion it is necessary to express clearly and concisely what the limitations of the study were. Once again, thanks for this insight. We have now attempted to address the study’s limitations in the revised manuscript; please refer to lines 396-408.

Reviewer 2 Report

1.     Suggest revision of sentence below, as the study questionnaires can only measure prevalence, they cannot measure prevalence AND demonstrate associations – only the statistical analysis can demonstrate associations.

Specifically, in this study, we used Sinhalese and Tamil translations of the NorVold Abuse Questionnaire and the Abuse Assessment Screen to measure prevalence and demonstrate associations between women’s experiences with obstetric violence in maternity care and lifetime and pregnancy-specific domestic violence.

2.     The “systematic review” referred to in line 97 is not actually a systematic review, was a literature review – suggest change in wording to reflect that.

3.     “Sample size was calculated using standard methods” (line 173) – please expand on what the standard methods are, as well as what the “design effect” referred to in the same sentence is.

4.     How was the randomisation carried out? (line 178)

5.     Some information/detail on the data collection instrument that the author developed is needed – other two instruments are described adequately (line 202)

Authors should be commended for expanding the research base on a very important topic.

Round 2

Reviewer 1 Report

It was a pleasure to review this second improved version of this manuscript dealing with obstetric violence in Sri Lanka.

The authors endeavored to comply with the recommendations given by the different reviewers.

I think this version is much improved over the first and could be published in its current form.

One suggestion for future posts is that when an improved version is made, the changes are reflected in red text to make it easier to review. The document I received was with all the text in black.

Thanks

Kind regards